# Electrohypersensitivity as a Newly Identified and Characterized Neurologic Pathological Disorder: How to Diagnose, Treat, and Prevent It

**DOI:** 10.3390/ijms21061915

**Published:** 2020-03-11

**Authors:** Dominique Belpomme, Philippe Irigaray

**Affiliations:** 1Association for Research Against Cancer (ARTAC), 57/59 rue de la Convention, 75015 Paris, France; philippei.artac@gmail.com; 2European Cancer and Environment Research Institute (ECERI), 1000 Brussels, Belgium; 3Department of Cancer Clinical Research, Paris V University Hospital, 75005 Paris, France

**Keywords:** electrohypersensibility, multiple chemical sensitivity, neurologic disease, oxidative stress, melatonin, O-myelin, inflammation, histamine, radiofrequency, extremely low frequency, electromagnetic fields

## Abstract

Since 2009, we built up a database which presently includes more than 2000 electrohypersensitivity (EHS) and/or multiple chemical sensitivity (MCS) self-reported cases. This database shows that EHS is associated in 30% of the cases with MCS, and that MCS precedes the occurrence of EHS in 37% of these EHS/MCS-associated cases. EHS and MCS can be characterized clinically by a similar symptomatic picture, and biologically by low-grade inflammation and an autoimmune response involving autoantibodies against O-myelin. Moreover, 80% of the patients with EHS present with one, two, or three detectable oxidative stress biomarkers in their peripheral blood, meaning that overall these patients present with a true objective somatic disorder. Moreover, by using ultrasonic cerebral tomosphygmography and transcranial Doppler ultrasonography, we showed that cases have a defect in the middle cerebral artery hemodynamics, and we localized a tissue pulsometric index deficiency in the capsulo-thalamic area of the temporal lobes, suggesting the involvement of the limbic system and the thalamus. Altogether, these data strongly suggest that EHS is a neurologic pathological disorder which can be diagnosed, treated, and prevented. Because EHS is becoming a new insidious worldwide plague involving millions of people, we ask the World Health Organization (WHO) to include EHS as a neurologic disorder in the international classification of diseases.

## 1. Introduction

The term electromagnetic hypersensitivity or electrohypersensitivity (EHS) was first proposed in 1991 by William Rea to identify the clinical condition of patients reporting health effects while being exposed to an electromagnetic field (EMF) [1]. This term was then used in 1997 in a report provided by a European group of experts for the European Commission to clinically describe this unusual pathology, which may imply EMF exposure [2].

In 2002, Santini et al. in France reported similar symptomatic intolerance in users of digital cellular phones and among people living near wireless communication base stations [3,4]. In 2004, because of the seemingly worldwide prevalence increase in EHS, the World Health Organization (WHO) organized an international scientific workshop in Prague to define and characterize EHS. Although not acknowledging EHS as being caused by EMF exposure, the Prague working group clearly defined EHS as “a phenomenon where individuals experience adverse health effects while using or being in the vicinity of devices emanating electric, magnetic, or electromagnetic fields” [5]. WHO then acknowledged EHS as an adverse health condition [6]. However, according to a previous 1996 International Program on Chemical Safety (IPCS)-sponsored conference in Berlin on multiple chemical sensibility (MCS) [7], it was recommended to qualify such unknown new pathological conditions under the term of “idiopathic environmental intolerance (IEI)”. Thus, following the Prague workshop, instead of using the term EHS, it was proposed to use the term “idiopathic environmental intolerance attributed to EMF (IEI-EMF)” to name this particular pathological condition, because of the lack of a proven causal link between EHS and EMF exposure, and no proven physiopathological mechanism linking EMF exposure with clinical symptoms.

That is indeed what WHO officially stated in its 2005 fact sheet 296 [6], indicating that “EHS resembles MCS, another disorder associated with low-level environmental exposure to chemicals …” and that because of “non-specific symptoms” and “no clear diagnostic criteria”, this “disabling condition” could not be diagnosed medically. In addition, in 2002 and 2013, WHO classified extremely low frequencies (ELF) and radiofrequencies (RF) respectively as possibly carcinogenic (group IIB), meaning that EMFs may cause cancer. This past scientific evolution is summarized in Table 1.

However, since the 2005 WHO statement on EHS and a more recent 2014 WHO report on mobile phone exposure and public health [8], much clinical and biological progress has been made in identifying and characterizing EHS, as summarized during the international scientific consensus meeting on EHS and MCS which we organized in May 2015 in Brussels at the Royal Belgium Academy of Medicine [9].

Because we suspected that EHS prevalence was increasing worldwide, since 2009, we constituted and maintained a database which was registered by the French Committee for the protection of persons (CPP), under the registration number 2017-A02706-47, as well as in the European Clinical *Trials* Database (*EudraCT*), under the registration number 2018-001056-36. This database presently includes more than 2000 EHS and/or MCS cases. All the patients included in this series gave their informed consent for clinical and biological research investigations. In addition, all these patients were anonymously registered in the database.

By querying this database, we showed for the first time that EHS is frequently associated with MCS [10], and that EHS and MCS are characterized by a common similar clinical picture which can be identified objectively by the detection of similar biomarkers in the peripheral blood and urine [10,11], and by similar pulsometric abnormalities in the brain [10,12]. Thus it finally appears that EHS and MCS could in fact be two etiopathogenic aspects of a unique pathological disorder [10]. We would like here to overview our original data and discuss the possibility that EHS is part of a true pathologic neurologic disorder resulting from a comprehensive physiopathologic mechanism, in common with MCS. We conclude that EHS—whatever its causal origin—is becoming a worldwide plague. Thus, as we showed that it can be diagnosed, treated medically, and eventually prevented, we ask WHO to include EHS in the international classification of diseases (ICD).

## 2. Demography

In a prospective study involving systematic face-to-face questionnaire-based interviews and clinical physical examinations of many patients constituting part of the database, we reported that EHS is a well-defined clinico-biological entity [10].

Table 2 presents the demographic data we obtained from the serial analysis of the first 726 consecutive cases included in the database. No children were included. Median and mean ages were 48 years for the EHS group, 48 and 47 years, respectively, for the MCS group, and 46 years for the EHS and MCS-associated group. Sex ratio shows a clear predominance of women among patients, reaching two-thirds in the EHS group and the MCS group, while it was three-quarters in the group of patients presenting with both disorders. This strongly suggests that women are genetically more susceptible than men to the environmental intolerance attributed to EMFs and/or chemicals.

## 3. Clinical Description

Table 3 presents the detailed symptomatic picture that we obtained during face-to-face interviews and clinical examinations for the groups of (1) EHS self-reported patients, (2) MCS self-reported patients, and (3) both disorder self-reported patients. Symptoms in patients with EHS were compared with those from a series of apparently healthy control subjects that showed no clinical evidence of EHS and/or MCS. As indicated in the table, EHS is characterized by the occurrence of neurologic symptoms including headache, tinnitus, hyperacusis, dizziness, balance disorder, superficial and/or deep sensibility abnormalities, fibromyalgia, vegetative nerve dysfunction, and reduced cognitive capability, including immediate memory loss, attention–concentration deficiency, and eventually tempo-spatial confusion. These symptoms were associated with chronic insomnia, fatigue, and depressive tendency, in addition to emotional lability and sometimes irritability. A major observation is that symptoms were repeatedly reported by the patients to occur each time they reported being exposed to presumably EMF sources, even of weak intensity, and to regress or even disappear after they left these presumed sources. With the exception of arthralgia and emotivity, which were observed at a similar frequency range in the control group, all clinical symptoms occurring in EHS patients were found to be significantly much more frequent than those in apparently normal controls.

Contrary to what was claimed from studies reporting clinical symptoms in EHS patients [2,5,6,13], these symptoms were not all subjective. In many cases, they were confirmed by family members; moreover, we were able to detect, at physical examination, a Romberg sign (objective posture test) in 5% of the cases and to observe the presence of cutaneous lesions in 16%. Overall, although many of these symptoms are considered as non-specific in the scientific literature, the general clinical picture resulting from their association and frequency strongly suggests that EHS can in fact be recognized and identified as a typical neurologic disorder as it is also the case for MCS and MCS-associated EHS.

Table 3 reveals that between EHS and MCS there is no statistically significant difference in types and frequencies of clinical symptoms for headache, myalgia and arthralgia, balance disorder, concentration/attention deficiency, emotivity and irritability, skin lesions and global body dysthermia, whereas dysesthesia, ear heat/otalgia, tinnitus, hyperacusis, dizziness, loss of immediate memory, insomnia and fatigue as well as depression tendency and suicidal ideation appear to be statistically more frequent in EHS than in MCS. Moreover, in the case of EHS associated with MCS, most of the symptoms—such as headache, dysesthesia, myalgia and arthralgia, tinnitus, and, above all, cognitive capability, including loss of immediate memory, concentration/attention deficiency, and tempo-spatial confusion—were found to be significantly more frequent than in EHS alone, suggesting that the presence of an additional chemical intolerance component to the intolerance attributed to EMF exposure is associated with a more severe pathology. This was especially the case for skin lesions which were found in 45% of the cases, as well as for physical and mental suffering and depressive tendency with underlying suicidal ideation in 40%.

Note that cutaneous lesions were more frequent on the superior members than on the inferior members of the patients, and more frequent on the hands, particularly on the hand which held the mobile phone (as exemplified in Figure 1A). Note also that the cutaneous lesions were not only more frequent in the group of patients with EHS- and MCS-associated disorders (45%) than in the group of patients with only EHS (16%), but also that they were more extensive and persistent in the cases of both associated disorders than in the case of EHS alone (Figure 1B).

These clinical observations strongly suggest that EHS and EHS/MCS are objective somatic disorders, which can neither be claimed as originating from some psychologic or psychiatric-related conditions, nor from nocebo effects [11] (see further).

## 4. Identification of Biomarkers

On the basis of previously published experimental data, we selected and identified several biomarkers in the peripheral blood and urine of EHS and/or MCS patients which can allow physicians to objectively characterize EHS and MCS as true somatic pathological disorders [10], discounting the hypothesis that EHS and MCS could be caused by a psychosomatic or nocebo-related process [11]. As indicated in Table 4, there is a similar increase in mean level values of low-grade inflammation-related biomarkers in the peripheral blood of patients with EHS, MCS, or both associated disorders. In addition, as far as frequency is concerned, we found hypersensitive C reactive protein (hs-CRP) to be increased in 12–15% of the cases, histamine in 30% to 40%, immunoglobulin E (IgE) in 20% to 25%, and heat-shock protein 27 (Hsp 27) and Hsp 70 in 12% to 30%. Note that, among these markers, IgE and histamine were found to be increased in patients with no proven allergy; thus, in the case of no associated allergy, histamine appears to be the most frequently involved biomarker in EHS, as well as in MCS, suggesting a low-grade inflammatory process is involved in the genesis of these two disorders. Consequently, it is believed that, as an inflammation mediator, histamine could play a major key contributing role in the physiopathologic mechanism which may account for the occurrence of the two disorders [11,14] (see further). Note also that, with the exception of Hsp 70, which was found to be less frequently increased in the MCS group, there was no significant difference between the three groups of patients for the percentage of patients with values above normal, nor any significant difference in mean increased values in comparison with normal values for all biomarkers in the three groups studied, meaning that EHS, MCS, and the association of both disorders may share a common low-grade inflammation-related physiopathologic mechanism for genesis.

Moreover, as indicated in Table 5, we were able to show that, in peripheral blood, there is an increase in S100B protein in 15–20% of the patients and an increase in nitrosative stress-related nitrotyrosine (NTT) in 8–30% in the EHS and/or MCS groups, suggesting that these biomarkers may reflect opening of the blood–brain barrier (BBB) in these patients, whatever the patient group considered, since it was shown that S100B protein [15,16] and nitrotyrosine [17,18,19,20] are markers associated with BBB opening. In addition, we detected the presence of autoantibodies against O-myelin in about 20% of all cases, whether EHS, MCS or both; meaning that an autoimmune response against the white matter of the nervous system occurres in patients; a finding that may in fact be the consequence of the occurrence of oxidative/nitrosative stress [10,21].

Moreover, more recently, we measured different oxidative and nitrosative stress-related biomarkers such as thiobarbituric acid reactive substances (TBARS), oxidized glutathione (GSSG), and NTT in the peripheral blood of EHS patients. As reported in Figure 2, we found that nearly 80% of EHS patients presented with an increase in oxidative/nitrosative stress-related biomarkers—more precisely, with only one of these three studied biomarkers in 43% of the patients, two of these biomarkers in 21% of them, and all three in 15% [22]. This clearly indicates that, in addition to low-grade inflammation and an anti-white matter autoimmune response, EHS can also be diagnosed by the presence of oxidative/nitrosative stress.

Finally, we also found that, in comparison with normal reference values, the 24-h urine 6-hydroxymelatonin (6-OHMS)/creatinine ratio was normal or significantly decreased in 88% of cases, while, due to a still unexplained process, it was significantly increased in 12%, whatever the group of patients considered. 6-OHMS is a melatonin metabolite. Decrease in melatonin production as a consequence of prolonged EMF exposure was experimentally evidenced both in animals and in humans [23,24]. However, since EMF exposure was also reported not to alter melatonin synthesis and secretion [25], an alternative plausible explanation could be that a decrease in the excretion of 6-OHMS in the urine may result from a decrease in melatonin metabolic bioavailability due to its increased intake and utilization of melatonin as a free radical scavenger [26,27]. This indeed could be the case in patients with a decrease in the 24-h urine 6-OHMS/creatinine ratio level, since, as shown above, most EHS patients present with oxidative/nitrosative stress. Thus, a decrease in 6-OHMS in the urine may in fact be a consequence of the antioxidative stress effect of this hormone rather than its decreased synthesis in the pineal gland. Consequently, such reduction in bioavailability may contribute not only to clinical sleep disturbance in these patients, but also to a decrease in host defense mechanisms, possibly putting these patients at risk of neurodegenerative disease and cancer [28,29].

Moreover, the development of oxidative/nitrosative stress-related autoimmune response may also contribute to weakening the putative protective health effect of the chaperone proteins Hsp 70 and Hsp 27 [30]. There is presently no clear explanation why, in 12% of the cases, instead of having a normal or significant decrease in the 24-h urine 6-OHMS/creatinine ratio, this ratio was significantly increased in comparison with normal control values. As indicated in Table 6, this may be due in some cases to an increased production of serotonin in the brain, since serotonin is a precursor neurotransmitter of melatonin.

As indicated in Table 6, changes in neurotransmitter levels revealed that EHS is associated with different abnormal neurotransmitter profiles, confirming EHS is a well-established new brain-related neurologic disorder.

## 5. Radiological Identification of Cerebral Neuro-Vascular Abnormalities

Classical brain imaging techniques including brain computerized tomography (CT) scans, brain magnetic resonance imaging (MRI), and brain angioscans are usually normal in EHS patients and in MCS or EHS/MCS patients, meaning that the normality of these investigations is not an argument against the diagnosis of these pathological disorders. Fortunately we have shown that development and use of other imaging techniques could be greatly helpful to increase our ability of objectively characterizing EHS and MCS, should they show abnormal function. In fact, as indicated in Table 7, by using transcranial Doppler ultrasound (TDU) in patients with EHS, we showed a decrease in the mean pulsatility index in one or both middle cerebral arteries, i.e., for one artery in 25% and 31% of the cases respectively for the right and left artery, and for both arteries in 50%. Moreover, for the dual EHS/MCS group of patients, it was for one artery in 20% of the cases and for both arteries in 50%. In addition, as far as resistance in the blood flow (BBF) is concerned, we found that, in EHS patients, BBF resistance was increased for one artery in 6.25% of the cases and for both arteries in 18.75%, while in EHS/MCS patients, it was 5–10% for one artery and 25% for both arteries. Note also that mean blood flow velocity was below normal values in 9.75% to 40% of the cases, while it was above normal values in 5% to 18.75%, depending on the EHS and EHS/MCS group considered (see Table 7). This suggests that, in EHS and/or MCS, BBF may be decreased in one or both of these brain arteries.

Moreover, by using ultrasonic cerebral tomosphygmography (UCTS) applied to the temporal lobes [12], we showed there is a significant decrease in mean pulsometric index in the middle cerebral artery-dependent tissue areas of these lobes, especially in the capsulo-thalamic area, which corresponds to the limbic system and the thalamus [12]. As exemplified in Figure 3, this tissue hypo-pulsation—mainly detected in the capsulo-thalamic area of these lobes—suggests that EHS and/or MCS are associated with a capillary BBF decrease in these two brain structures, thus leading to the hypothesis that they may be associated with some vascular and/or neuronal dysfunction [10,11,12]. Although these abnormalities are not specific, since they may be similar to those found in Alzheimer’s disease and other neurodegenerative disorders, we recently confirmed that UCTS could presently be one of the most accurate imaging techniques to be used to diagnose EHS and/or MCS and to follow objectively treated patients [12].

It appears, however, that these brain abnormalities are not restricted to the limbic system and the thalamus, since, by using TDU as indicated above, we showed that, in EHS and/or MCS patients, BBF in the middle cerebral arteries may be abnormal. Moreover, by using functional MRI (fMRI) in EHS patients exposed chronically to extremely low-frequency (ELF) radiation, regional BBF changes were also reported by Heuser and Heuser, but mainly in the frontal lobes, as an abnormal default mode network (DMN) (particularly as hyper-connectivity of this DMN), in association with a decrease in cerebral BBF and metabolic processes in the two so far individualized fragment hyper-connected components [31]. For example, in Figure 4, abnormal DMN is represented with fragmented hyper-connectivity of the anterior component and posterior component, which may lead to decreased BBF and/or metabolism in the bi-frontal lobes.

## 6. Diagnostic Criteria

On the basis of the above clinical, biological, and radiological reported investigations, it appears that there is presently sufficient comprehensive and relevant data allowing the objective characterization and identification of EHS as a well-defined new neurologic pathological disorder. As a result, patients who self-report that they suffer from EHS should be investigated utilizing presently available objective tests, including the use of the above-reported blood and urine biomarkers and imaging techniques.

At a clinical level, isolated symptoms such as headache, tinnitus, dizziness, or cognitive defects, although they may be referred by the patients as being due to EMF or chemical exposure, are indeed not sufficient for the diagnosis to be made, as they may reflect another pathology. Clinical arguments for EHS could nevertheless be the following: (1) absence of known pathology accounting for the observed clinical symptoms; (2) characteristic association of symptoms such as those we identified, with the association of headache, tinnitus, hyperacusis, dizziness, loss of immediate memory, and attention/concentration deficiency being the most characteristic and reproducible; (3) reproducibility of symptoms under the said influence of EMFs; (4) regression or disappearance of symptoms in the case of said EMF avoidance; (5) finally and most importantly, the association with MCS. As we showed that MCS is associated with EHS in 30% of the cases, and as MCS was well defined during a 1999 international consensus meeting [32], this latter association may in fact be the best clinical criterion for the diagnosis of EHS.

However, because many of these clinical criteria are subjective, they are not sufficient to objectively prove the disease and, thus, establish the diagnosis. Among biological markers, histamine in the blood is presently the best available marker in the case of no associated allergy and the easiest to measure routinely in medical practice. Moreover, detection in the blood of an increase in protein S100B and oxidative/nitrosative stress-related biomarkers such as GSSG and NTT may also be objective contributing elements for the diagnosis. Note, however, that, in 30% of the cases, there were no positive detectable biomarkers in the blood; thus, in addition to the availability of clinical criteria, the EHS diagnosis could be made by using imaging techniques, such as TDU, fMRI, and, if possible, UCTS. Overall, by using this approach, we were able to objectively diagnose EHS in about 90% of EHS self-reported patients.

## 7. Treatment and Prognostic Evolution

There is, at the moment, no recognized standardized treatment of EHS. There are, however, some treatments that could be indicated, on the basis of biological investigations. We showed, for example, that patients with EHS present frequently with a profound deficit in vitamins and trace elements, especially in vitamin D and zinc, which should be corrected [10,11,22]. Anti-histaminics should also be used in the case of increased histamine in the blood. Furthermore, antioxidants such as glutathione and, more specifically, anti-nitrosative medications should also be used in case of oxidative/nitrosative stress. Moreover, as exemplified in Figure 5, we showed that natural products such as fermented papaya preparation (FPP) and ginkgo biloba can restore brain pulsatility in the various middle cerebral artery-dependent tissue areas of temporal lobes, thereby improving brain hemodynamics and, consequently, brain oxygenation [33]. Since FPP was shown to possess some antioxidant, anti-inflammation, and immune-modulating properties [34,35,36], we recommend the use of this widely available natural product.

In the case of no treatment and no protection against environmental stressors such as EMF and multiple chemicals, EHS may evolve toward some neurodegenerative and psychiatric disorders, possibly including some seemingly Alzheimer’s disease-related states. However, in treating and protecting patients as soon as possible, we never observed the occurrence of true Alzheimer’s disease in any patient included in the database. By contrast, regression and even disappearance of symptoms of intolerance may occur after treatment and protection of patients. However, in our experience and to our knowledge, hypersensitivity to EMF and/or MCS-related chemical sensitivity never disappears, meaning – unlike symptomatic intolerance – EHS and MCS appear to be associated with some irreversible neurologic pathological state, requiring strong and persistent prevention. So, contrary to some recent claims, we believe these disorders cannot be merely reduced to some type of functional impairment.

## 8. Proposed Physiopathological Mechanism

In its 2005 official statement on EHS, WHO indicated there is “no scientific basis to link EHS symptoms to EMF exposure” meaning there is no accepted physiopathological mechanism to link environmental cause to disease. This is no longer the case. The basic low-grade inflammation and oxidative/nitrosative stress-related states we showed in EHS patients [10,11,22] are remarkable since they confirm the detrimental health effects of (1) non-thermal or weak thermal non-ionizing radiation, which were proven experimentally in animals [37,38,39] and in humans [11] exposed to different environmental stressors including ELF and RF EMFs, and (2) multiple man-made environmental chemicals [40,41,42], especially in the brain [43,44].

Figure 6 summarizes the different steps of the model we have so far been able to construct from the presently available published data, including our own. On the basis of the inflammation and oxidative/nitrosative stress processes which we evidenced in EHS and/or MCS patients, this model accounts for the mechanisms via which physiopathological effects could take place in the brain and, consequently, how EHS and/or MCS genesis can occur.

In a first step, there could be an initial local inflammatory response to environmental stressors, whatever they may be. Resident microglia cells, astrocytes, and mastocytes could be the first cells in the brain locally involved in the inflammatory process, releasing inflammatory mediators such as histamine. On the basis of our data [10,11,12,22,33], it is speculated that histamine is a key mediator contributing to the induction of oxidative/nitrosative stress and, consequently, to cerebral hypoperfusion, thereby leading to some local cerebral hypoxia.

In a second step, amplification of inflammation could occur, including oxidative/nitrosative stress-related BBB disruption, allowing transmigration of circulating inflammatory cells from the blood to the brain. Finally, neuroinflammation in the brain would occur, mainly involving the capsulo-thalamic area of temporal lobes, i.e., the limbic system and the thalamus.

The major interest of this comprehensive physiopathological model is that it can explain the main clinical symptoms occurring in EHS and/or MCS patients, since the limbic system involvement may account for both the emotional and cognitive pathological alterations (in particular memory loss), while the thalamic involvement may explain sensibility-related abnormalities, both superficial and deep. Naturally, the possible extension of neuroinflammation into the frontal lobes and possibly into the hypothalamus [45] may, in addition, account for the other associated clinical symptoms.

## 9. Etiopathogenesis and Prevention

The causal origin of EHS is still debated, and the present current institutional message is that there is no proof that EHS genesis is causally related to EMF exposure. There is, however, great confusion in the present scientific literature in addressing this problem, since there is presently no clear distinction between the cause of clinical symptoms occurrence in EHS patients, i.e., after EHS has already occurred, and the environmental causal origin of EHS itself. In fact, as reported in Table 8, by querying the database and analyzing retrospectively previous exposure to EMFs and/or chemicals in EHS- and EHS/MCS-bearing patients, we found there are presently several direct and indirect arguments which strongly suggest that EMF exposure and even chemicals may cause or contribute to cause EHS.

Moreover, a further distinction should be made between the general term of intolerance, which refers to the clinical symptoms and/or the biological abnormalities occurring in a particular environmental situation, and the term hypersensitivity, which should in fact be defined as a particular endogenous physiopathological state characterized by a decrease in the environmental tolerance threshold to such a critical point that patients become intolerant to low-dose stressors. Such a distinction is already made in medicine as, for example, the individualization of atopy in allergic patients.

Thus, if we agree on the distinction between the concept of intolerance and that of EHS, EHS should be characterized by definition as a particular decrease in the intolerance threshold according to which patients become intolerant to low-dose-intensity EMF exposure, while MCS (as already indicated by the MCS consensus meeting report in 1999 in Atlanta) was defined by a similar physiopathological state in which patients become intolerant to low-dose multiple chemicals [32]. This distinction may explain why most studies using provocation tests aiming to reproduce the clinical symptoms which may occur under EMF exposure in EHS self-reported patients report negative findings. Indeed, these negative results may in fact be due to different, unacceptable scientific flaws: (1) the lack of objective inclusion criteria, because objective biomarkers were not used to define EHS in so-called EHS-self reported patients; (2) EHS patients may be sensitive to certain frequencies and not necessarily to others; (3) duration of exposure was generally too short and assessment too early; (4) association with MCS was not considered; (5) as reported above, EHS patients have cognitive defects and, thus, can make mistakes in distinguishing EMF exposure from sham exposure; (6) and above all, patients may respond positively in the case of sham exposure because of a decrease in environmental tolerance threshold, as well as because of psychologic conditioning from their past history of suffering.

Hence, on this basis, and because of the experimental evidence provided by studies in animals [37,38,39,43,44] and in humans [11,14,23,24] have shown the detrimental impact of EMF on health we believe, there is presently no sufficiently robust scientific data to refute a role of EMF exposure in inducing the previously described clinical symptoms and biological alterations in EHS patients.

Therefore, the causal origin of EHS should be established with a different scientific approach. RF and ELF were found to cause persistent adverse biological effects not only in animals [46,47] but also in plants [48,49] and microorganisms [50]. Here too, such observations certainly dismiss the hypothesis of a nocebo effect as the initial cause of EHS. In fact, the inflammation and oxidative/nitrosative states we showed in EHS patient are remarkable since they confirm the data obtained experimentally in animals exposed to these two types of non-ionizing frequencies [37,38,39], especially in the brain [43,44]. Furthermore, the limbic system-associated capsulo-thalamic abnormalities that we showed to characterize these patients [12,33] may likely correspond to the hippocampal neuronal alterations caused by EMF exposure in rats [51,52,53].

We therefore consider that the biological effects we observed in EHS patients may be due to both the pulsed and the polarized characteristics of man-made EMF emitted by electric or wireless technologies, as opposed to terrestrial non-polarized and continuously emitted natural EMFs [54,55,56].

In addition, as indicated in Table 9, we showed that, in 30% of the EHS cases, EHS was associated with MCS, with MCS preceding the occurrence of EHS in 37% of these EHS/MCS-associated cases; meaning that in this group of patients, EHS evolved toward MCS in 63% of the cases. As reported in Table 8, we thus speculate that man-made environmental chemicals may also be causally involved in EHS genesis in around 11% of the cases.

These various considerations should not be neglected, since to avoid risks, knowledge of them could lead to protective measures in EHS and/or MCS patients. Such measures should include as much as possible EMF and chemical avoidance, use of anti-EMF clothes, and earthing-related electric charge detoxication. In addition, public preventive measures for the most vulnerable people—particularly pregnant women, infants, children, and adolescents—should be taken by limiting or even totally avoiding the use of wireless technology in these conditions. Such protective measures should also be taken and carried out in vulnerable patients, i.e., in cardiac patients with pacemakers, in patients with auditive prothesis, and in patients with neurodegenerative diseases.

## 10. The Worldwide Health Plague

Another argument incriminating the role of new wireless technology and possibly man-made chemicals introduced in the environment [57,58] is that, as indicated in Table 10, the increase in EHS prevalence is not restricted to a single country but is presently a worldwide plague, which started as soon as these industrial technologies became widespread. Prevalence of EHS occurrence is estimated to range from 0.7% to 13.3%, mainly affecting about 3% to 5% of the population in many countries (Table 10), meaning that millions of people may in fact be affected by EHS worldwide.

Furthermore, although these reported EHS prevalence figures are only estimations, not critically evaluated due to a lack of objective criteria to clearly define EHS, it is possible—as speculated in Figure 7—that the EHS prevalence will continue to grow in the future, in as much as the manufacture of wireless technology and industrial chemicals will continue to develop.

## 11. Conclusions

In summary, we showed that there are presently sufficient clinical, biological, and radiological data for EHS to be acknowledged as a well-defined, objectively identified, and characterized pathological neurologic disorder. As a result, patients who self-report they suffer from EHS should be diagnosed and treated on the basis of presently available biological tests, including the detection of peripheral blood and urine biomarkers and the use of imaging techniques such as fMRI, TDU, and, when possible, UCTS. Moreover, because we showed for the first time that EHS is frequently associated with MCS and that both clinico-biological entities may be associated with a common physiopathological mechanism for genesis, it clearly appears that they can be identified as a unique neurologic pathological syndrome, whatever their causal origin. Moreover; as it was shown that MCS genesis may be attributed to toxic chemical exposure, and EHS genesis to potentially excessive EMF and/or chemical exposure; protective measures against these two environmental stressors should be taken.

Whatever its causal origin and mechanism of action, EHS should therefore be from now on recognized as a new identified and characterized neurological pathological disorder. As it is already a real health plague potentially involving millions of people worldwide it should be acknowledged by WHO, and thus be included in the WHO ICD. As stated during the international scientific consensus meeting on EHS and MCS that we have organized in 2015 in Brussels, scientists unanimously asked WHO to urgently assume its responsibilities, by classifying EHS and MCS as separate codes in the ICD; so as to increase scientific awareness of these two pathological entities in the medical community and the general public, and to foster research and train medical practitioners to efficiently diagnose, treat, and prevent EHS and MCS–, which in fact constitute a unique, well-defined, and identifiable new neurologic disease.

## Figures and Tables

**Figure 1 ijms-21-01915-f001:**
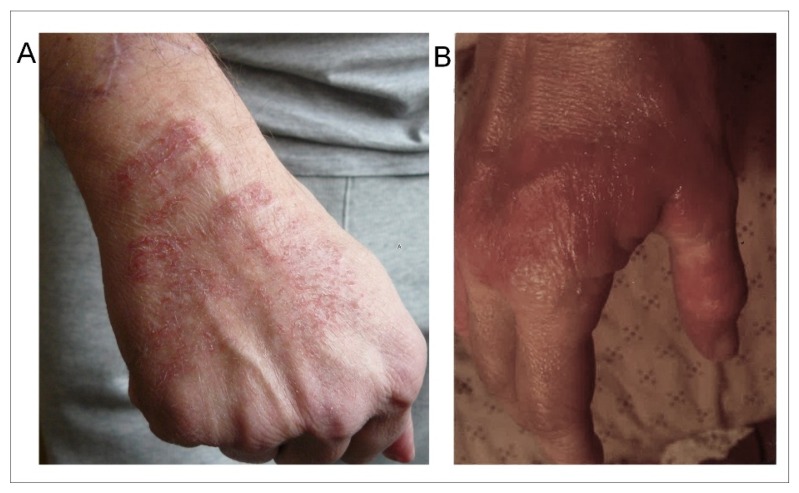
Examples of skin lesions observed on the hand of an EHS-bearing patient (**A**) and of an EHS/MCS-bearing patient (**B**). (Photographs are issued from the database).

**Figure 2 ijms-21-01915-f002:**
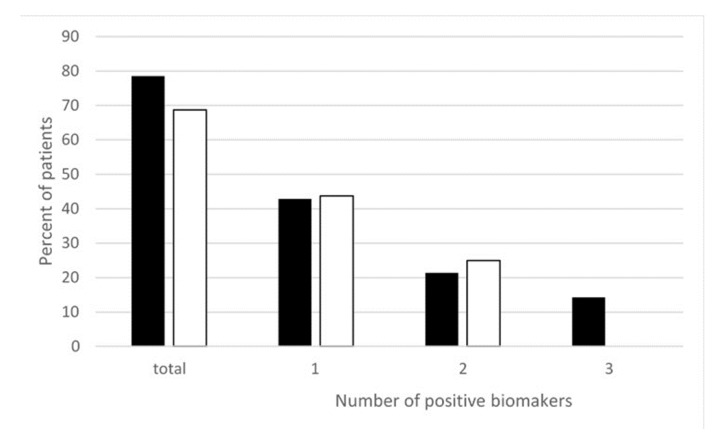
Percentage of EHS self-reported patients having positive thiobarbituric acid reactive substances (TBARS), oxidized glutathione (GSSG), and/or NTT oxidative stress biomarkers measured in the peripheral blood, according to Reference [22]. ■ Corresponds to NTT, TBARS, and GSSG, i.e., all three biomarkers measured in 14 of the 32 included patients. ☐ Corresponds to TBARS and GSSG analyzed in all 32 included patients. “Positive” biomarkers correspond to patients having one, two, or three markers with levels above the upper normal limits, and “total” corresponds to patients having at least one positive biomarkers, i.e., having one, two, or possibly three positive biomarkers.

**Figure 3 ijms-21-01915-f003:**
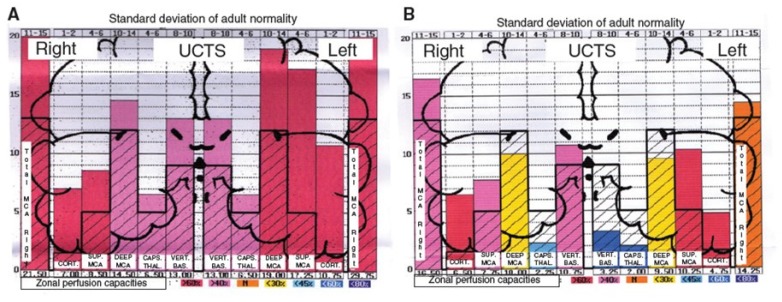
Examples of diagrams obtained from the database by using ultrasonic cerebral tomosphygmography (UCTS), exploring the global centimetric ultrasound tissue pulsatility in the two temporal lobes of a normal subject (**A**) and of an EHS self-reported patient (**B**), according to References [11,12]. Measurements are expressed as pulsometric index (PI). Note that, in A and B, mean values of PI in each explored area are recorded from the cortex to the internal part of each temporal lobe (i.e., from left to right for the right lobe, and from right to left for the left lobe). In addition, note that, in A (normal subject), all values are over the median normal PI values, whereas, in B (EHS self-reported patient), values in the so called capsulo-thalamic areas (the fifth and the second column for the right and left temporal lobes, respectively) are significantly under the median normal values, suggesting that the limbic system and the thalamus in each temporal lobe may be involved in EHS, as exemplified in this patient.

**Figure 4 ijms-21-01915-f004:**
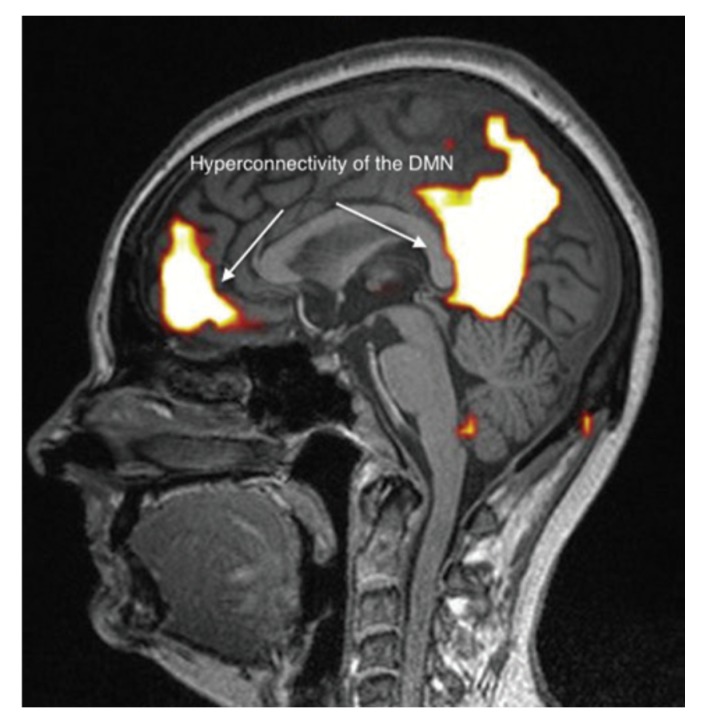
Abnormal functional MRI brain scan in patients complaining of EHS after long-term exposure to EMF, according to Reference [31].

**Figure 5 ijms-21-01915-f005:**
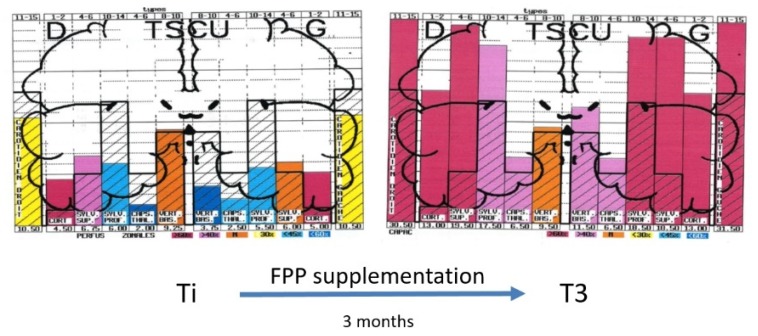
Example of diagrams obtained from the database by using UCTS exploring the global centimetric ultrasound pulsatility in the two temporal lobes of an EHS subject at inclusion (Ti) and three months later (T3) after fermented papaya preparation (FPP) supplementation (9 g per day in two divided doses), according to Reference [33].

**Figure 6 ijms-21-01915-f006:**
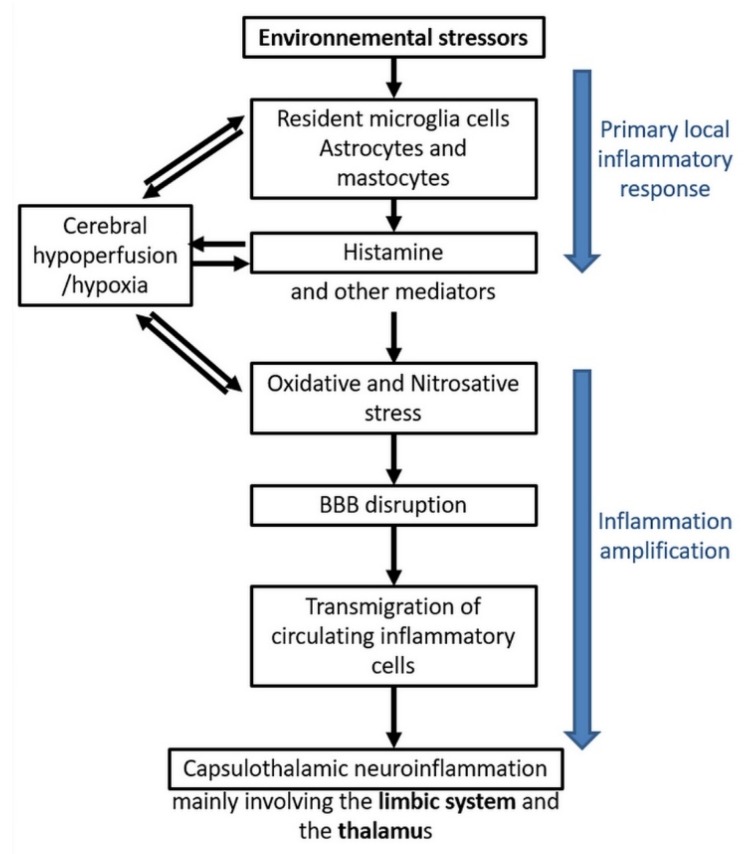
EHS/MCS physiopathological model based on low-grade neuroinflammation and oxidative/nitrosative stress-induced blood–brain barrier disruption, according to Reference [10].

**Figure 7 ijms-21-01915-f007:**
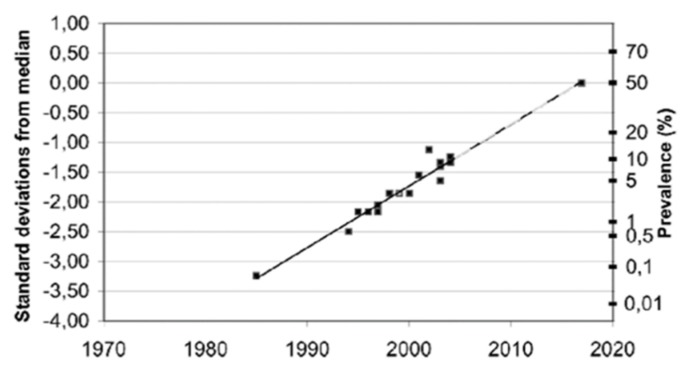
Estimated prevalence (%) of people around the world who consider themselves to be electrohypersensitive, plotted over time in a normal distribution graph, according to Reference [73].

**Table 1 ijms-21-01915-t001:** Electrohypersensitivity (EHS)/multiple chemical sensitivity (MCS) and cancer statements including those of the World Health Organization (WHO) or on behalf of WHO. COST—European action for co-operation in the field of science and technological research on biological effects of electromagnetic fields; EMF—electromagnetic field; IARC—international agency for research on cancer.

1996	Berlin: WHO-sponsored workshop; MCS classified as idiopathic environmental intolerance (IEI)
1997	Stockholm: Possible health implication of electromagnetic field exposure; a report prepared by a European group of experts for the European Commission
1998	Austria: COST 244 bis international workshop on EHS
1998	Atlanta (US): MCS 1999 consensus meeting
2002	IARC: Extremely low frequency (ELF) EMFs classified as possibly carcinogenic (Group IIB)
2004	Prague: WHO workshop; identification of idiopathic environmental intolerance attributed to EMF
2005	WHO: WHO fact sheet n° 292 aiming at defining EHS
2013	IARC: Radiofrequency (RF) EMFs classified as possibly carcinogenic (Group IIB)
2015	Brussels: Fourth Paris Appeal Colloquium; a focus on electromagnetic fields and EHS

**Table 2 ijms-21-01915-t002:** Age and sex ratio in EHS and/or MCS self-reported patients, according to Reference [10].

Demographic Data	EHS	MCS	EHS/MCS
*n* (%)	521 (71.7%)	52 (7.1%)	154 (21.2%)
Age (mean ± SD)	48.2 ± 12.9	48.5 ± 10.3	46.7 ± 11.2
Age (median (range))	48 (16–83)	47 (31–70)	46 (22–76)
Sex ratio (women/men)	344/177	34/18	117/37
Female (%)	66	65	76

**Table 3 ijms-21-01915-t003:** Clinical symptoms in EHS self-reported patients in comparison with those in normal controls and in comparison with those in MCS and EHS/MCS self-reported patients *, according to Reference [11].

Clinical Symptoms	EHS (%)	Normal Controls (%)	*p* **	MCS (%)	*p* ***	EHS/MCS (%)	*p* ****
Headache	88	0	<0.0001	80	0.122	96	0.065
Dysesthesia	82	0	<0.0001	67	0.0149	96	0.002
Myalgia	48	6	<0.0001	48	1	76	<0.0001
Arthralgia	30	18	0.067	24	0.611	56	<0.001
Ear heat/otalgia	70	0	<0.0001	16	<0.0001	90	<0.001
Tinnitus	60	6	<0.0001	35	<0.001	88	<0.0001
Hyperacusis	40	6	<0.0001	20	<0.001	52	0.118
Dizziness	70	0	<0.0001	52	0.0137	68	0.878
Balance disorder	42	0	<0.0001	40	0.885	52	0.202
Concentration/attention deficiency	76	0	<0.0001	67	0.210	88	0.041
Loss of immediate memory	70	6	<0.0001	56	0.040	84	0.028
Confusion	8	0	0.007	0	0.0038	20	0.023
Fatigue	88	12	<0.0001	72	0.0047	94	0.216
Insomnia	74	6	<0.0001	47	<0.0001	92	0.001
Depression tendency	60	0	<0.0001	29	<0.0001	76	0.022
Suicidal ideation	20	0	<0.0001	9	0.027	40	0.003
Transitory cardiovascular abnormalities	50	0	<0.0001	36	0.046	56	0.479
Ocular deficiency	48	0	<0.0001	43	0.478	56	0.322
Anxiety/panic	38	0	<0.0001	19	0.003	28	0.176
Emotivity	20	12	0.176	16	0.461	20	1
Irritability	24	6	<0.001	14	0.071	24	1
Skin lesions	16	0	<0.0001	14	0.692	45	<0.0001
Global body dysthermia	14	0	<0.0001	6	0.236	8	0.258

* These data result from the clinical analysis of 150 consecutive clinically evaluable cases issued from the database including an already published series of EHS and/or MCS patients who were investigated for biological markers [10]. Symptoms in EHS self-reported patients were compared with symptoms obtained from a series of 50 apparently normal subjects used as controls. These symptoms were also compared to those occurring in MCS and EHS/MCS self-reported patients. Percentage of patients with symptoms were compared by using the chi-square independence test. ** Statistical difference between EHS self-reported patients and normal controls. *** Statistical difference between EHS self-reported patients and MCS self-reported patients. **** Statistical difference between EHS self-reported patients and EHS/MCS self-reported patients.

**Table 4 ijms-21-01915-t004:** Increase in low-grade inflammation-related biomarker mean blood level values in the peripheral blood of patients with EHS and/or MCS, according to References [9,10]. SE—standard error; hs-CRP—hypersensitive C reactive protein; IgE—immunoglobulin E; Hsp—heat-shock protein.

	Patient Groups							
Marker Normal Values	EHS Mean ± SE	Above Normal (%)	MCS Mean ± SE	Above Normal (%)	*p* *	EHS/MCS Mean ± SE	Above Normal (%)	*p* **
hs-CRP < 3 mg/L	10.3 ± 1.9	15	5.3 ± 1.7	12	0.50	6.9 ± 1.7	14.3	0.36
Histamine < 10 nmol/L	13.6 ± 0.2	37	23.5 ± 4.5	33	0.91	13.6 ± 0.4	41.5	0.52
IgE < 100 UI/mL	329.5 ± 43.9	22	150.9 ± 18.3	20	0.23	385 ± 70	24.7	0.53
Hsp 70 < 5 ng/mL	8.2 ± 0.2	18.7	5.9 ± 0.5	12	0.03	8 ± 0.3	25.4	0.72
Hsp 27 < 5 ng/mL	7.3 ± 0.2	25.8	6.8 ± 0.1	6 ***	0.59	7.2 ± 0.3	31.8	0.56

* Comparison between the EHS and MCS groups of patients for marker mean level values was done using the two-tailed *t*-test. Except for Hsp 70, there is no statistically significant difference between EHS and MCS patients for increased mean level values of the different biomarkers analyzed, suggesting that EHS and MCS share a common physiopathological mechanism for genesis. ** Comparison between the EHS and EHS/MCS groups of patients by using the two-tailed *t*-test. There is no statistically significant difference between EHS and EHS/MCS patients for increased mean level values of the different biomarkers analyzed. *** With the exception of MCS, for which there is a statistically significantly lower frequency percentage value for Hsp 27, the frequency percentage values obtained in EHS and EHS/MCS for all the other investigated parameters do not differ significantly on the basis of the chi-square independence test.

**Table 5 ijms-21-01915-t005:** Increase in mean blood level values of peripheral blood S100B protein, nitrotyrosine (NTT), and O-myelin autoantibodies in EHS and/or MCS patients, according to References [10,11].

	Patient Groups							
Markers Normal Values	EHS Mean ± SE	Above Normal (%)	MCS Mean ± SE	Above Normal (%)	*p* *	EHS/MCS Mean ± SE	Above Normal (%)	*p* **
S100B < 0.105 µg/L	0.20 ± 0.03	14.7	0.25 ± 0.05	21.15	0.56	0.17 ± 0.03	19.7	0.69
NTT * > 0.9 µg/ml	1.36 ± 0.12	29.7	1.26 ± 0.13	8	0.85	1.40 ± 0.12	28.9	0.86
O-myelin (qualitative test)	Positive	22.8	Positive	13.6	_	Positive	23.6	_

* Comparison between the EHS and MCS groups of patients using the two-tailed *t*-test. There is no statistically significant difference between the two groups of EHS and MCS patients for increased mean level values of the two different biomarkers analyzed, suggesting that EHS and MCS share a common physiopathological mechanism for genesis. ** Comparison between the EHS and EHS/MCS groups of patients using the two-tailed *t*-test. There is no statistically significant difference between EHS and EHS/MCS patients for increased mean level values of the different biomarkers analyzed, suggesting here too that EHS and MCS share a common physiopathological mechanism for genesis.

**Table 6 ijms-21-01915-t006:** Preliminary unpublished data based on the measurement of neurotransmitters and their metabolites in the urine of 42 EHS-bearing patients. 3-4 DOPAC—3,4-Dihydroxyphenylacetic acid.

Neurotransmitters	Patients	%
Dopamine increase	17/42	31
3-4 DOPAC decrease	18/42	43
Noradrenaline increase	11/42	26
Adrenaline increase	8/42	19
Adrenaline decrease	12/42	22
Serotonin increase	4/42	9.5
Serotonin decrease	5/42	12

**Table 7 ijms-21-01915-t007:** Results of resistance index, pulsatility index, and mean flow velocity in comparison with normal values in the right and left middle cerebral arteries using transcranial Doppler ultrasound in 32 EHS cases and 20 EHS/MCS cases (unpublished data).

		**EHS *n* = 32**							
	**Normal Value**	**Mean ± SE**		**Below Normal (%)**			**Above Normal (%)**		
	**Right and Left**	**Right**	**Left**	**Right Only**	**Left Only**	**Both**	**Right Only**	**Left Only**	**Both**
Resistance index	<0.75	0.62 ± 0.03	0.65 ± 0.04	_	_	_	6.25	6.25	18.75
Pulsatility index	>0.60	0.55 ± 0.02	0.55 ± 0.03	25	31.25	50	_	_	_
Mean flow velocity	62 ± 12	59.56 ± 5.98	61.35 ± 5.27	9.75	9.75	31.25	3.12	9.25	18.75
		**EHS/MCS *n* = 20**							
	**Normal values**	**Mean ± SE**		**Below Normal (%)**			**Above Normal (%)**		
	**Right and Left**	**Right**	**Left**	**Right only**	**Left only**	**Both**	**Right only**	**Left only**	**Both**
Resistance index	<0.75	0.79 ± 0.09	0.64 ± 0.04	_	_	_	5	10	25
Pulsatility index	>0.60	0.48 ± 0.03	0.61 ± 0.02	20	0	65	_	_	_
Mean flow velocity	62 ± 12	53.03 ± 9.09	51.77 ± 7.63	20	20	40	10	10	5

**Table 8 ijms-21-01915-t008:** Clinical analysis of self-reported excessive presumed EMF and chemical exposure preceding the occurrence of electrohypersensibility (unpublished data). DECT—digital enhanced cordless telecommunications; RF—radiofrequency; ELF—extremely low frequency.

Sources	EHS (%)	Frequency Bands
Mobile phone	37	RF
Mobile phone/DECT	8	
DECT	7	
Cathode-ray screen	9	
WiFi	16	
Relay antenna towers	3	
Energy-saving lamps/mobile phone *	1.4	RF and ELF
High-voltage power lines	2.7	ELF
Power transformer	1.7	
Railway	0.8	
Chemicals	11	
Idiopathic **	2.4	

* Presumed excessive source exposure concern both low frequencies (LF) and radiofrequencies (RF); ** possible genetic susceptibility.

**Table 9 ijms-21-01915-t009:** Percentage of MCS patients who later suffered from EHS and vice versa.

	Total EHS/MCS Patients	Total EHS Patients Including EHS/MCS Patients *
Percentage of MCS patients that later suffered from EHS	37	11
Percent of EHS patients that later suffered from MCS	63	19

* EHS/MCS patients represent 30% of the total number of EHS patients.

**Table 10 ijms-21-01915-t010:** Estimated prevalence of people with self-reported EHS in different worldwide countries. USA—United States of America.

Country	Date	Sample Size	People Contribution Rate (%)	Estimated % of People with EHS	References
Sweden	1997	15,000 (19–80) *	73	1.5	Hillert et al., 2002 [59]
Sweden	2010	3406	40	2.7	Palmquist et al., 2014 [60]
Swiss	2004	2048 (>14) *	55.1	5	Schreier et al., 2006 [61]
Swiss	2008	1122 (30–60) *	37	8.6	Roosli et al., 2010 [62]
Swiss	2009	1122 (30–60) *	37	7.7	Roosli et al., 2010 [62]
Germany	2004	30,047	58.6	10.3	Blettner et al., 2009 [63]
Germany	2004	30,047	58.4	8.7	Kowall et al., 2012 [64]
Germany	2006	30,047	58.4	7.2	Kowall et al., 2012 [64]
USA (California)	1998	2072	58.3	3.2	Levallois et al., 2002 [65]
Finland	2002	6121	40.8	0.7	Korpinen et al., 2009 [66]
Great Britain	Before 2007	3633	18.2	4	Eltiti et al., 2007 [67]
Taiwan	2007	1251	11.5	13.3	Tseng et al., 2011 [68]
Austria	Before 2008	460	88	3.5	Schröttner and Leitgeb, 2008 [69]
Japan	Before 2009	2472	62.3	1.2	Furubayashi et al., 2009 [70]
Holland	2011	5789	39.6	3.5	Batiatsas et al., 2014 [71]
Holland	Before 2013	1009	60	7	Vabn Dongen et al., 2014 [72]

* When precised, age intervals of included patients are indicated in brackets.

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
