# Peer review of "Electrohypersensitivity as a Newly Identified and Characterized Neurologic Pathological Disorder: How to Diagnose, Treat, and Prevent It"

_ijms, 2020, doi:10.3390/ijms21061915_

Round 1

Reviewer 1 Report

Review

Belpomme D. and Rigaray P.  presented a very interesting  review, and I recommend it  for the publication. I have only a few suggestions for the improvement

A list of abbreviations I suggest the change in the logics of the presentation. Epidemiology, here world health plague Demography Pathophysiology: combine physio-pathological mechanisms and ethiopathogenesis Diagnostic criteria Biochemistry imaging techniques Prevention and treatment can be in one section

In this way presented the knowledge will be shaped as in the majorityof textbooks.

Do not put the brake after the title of the table. The table should follow the title and  the table can be continued to a new page. Please check and put comas after the words such as Moreover,…However, In summary, Consult with a native English speaking person. Please write consistently 24H not 24 H, in my view 24h is better Footnote to Fig 1. NTT, not MTT- probably Page 10. Fig 3, I think complaining what, no preposition with Page 14 generally too short; assessment too early I would put the number of the ICD code- iCD-11, probably? Treatment and evolution. I thein the word evolution is not suitable here. Please check if the references are referred correctly in this section, Vitamin D and Zink:  are the references correct? Page 13. Please consider the sentence The casual origin of symptoms in EHS patients... On my opinion this sentence is difficult to understand.

All in all  the manuscript is very good a but it needs a small brush-up.

Reviewer 2 Report

  • How did individuals end up in the database? Over what period of time were they enrolled?
  • I believe Table 2 can be improved. The first column needs a label. I believe the first row would be better labeled n(%). Then % does not need to be repeated. Why is there no % for MCS? The second row would be better labeled as Age (mean±SD), y. The third row would be better labeled as Age (median [range]),y. The third row would be better labeled as Sex ratio (women/men). Then W & M don’t need to be repeated. The fourth row could be labeled Female (%). Then % would not have to be repeated.
  • The first column of Table 3 needs to be labeled. For consistency, II would label the second and third columns as EHS(%) & Normal controls(%). I would label the 5th & 7th columns as MCS(5) & EHS/MCS(%). That way the % sign does not have to be repeated in subsequent rows.
  • Were skin lesions more prevalent in one location compare to another,
  • Were the biomarkers selected a-priori based on some rationale, or were they among many tested?
  • Tables 5,7&8 would benefit from changes similar to those suggested for earlier tables.
  • The figures are nicely contributory, but some would benefit from modification. To be consistent I would capitalize the first word of all cell text (eg, Histamine).
  • I may have missed it, but I did not notice any reference to IRB approval or informed consent.
